# Enhanced oxidative stress and damage in glycated erythrocytes

**Chloé Turpin[1], Aurélie Catan[2], Alexis Guerin-Dubourg[2], Xavier Debussche[3,4], Susana B. Bravo[5], Ezequiel Álvarez[5], Jean Van Den Elsen[6], Olivier Meilhac[1,7], Philippe Rondeau[1]\*, Emmanuel Bourdon[1]\***

**1** Université de La Réunion, INSERM, UMR 1188 Diabète athérothombose Thérapies Réunion Océan Indien (DéTROI), Saint-Denis de La Réunion, France, **2** Centre hospitalier Ouest Réunion, Saint-Paul, France, **3** CHU de La Réunion, Service d'endocrinologie, Saint Denis, France, **4** Centre d'Investigations Cliniques 1410 INSERM, Reunion University Hospital, Saint-Pierre, Réunion, France, **5** Proteomic Unit and Instituto de Investigación Sanitaria de Santiago de Compostela (IDIS), Complexo Hospitalario Universitario de Santiago de Compostela (CHUS), SERGAS, Santiago de Compostela, CIBERCV, Madrid, Spain, **6** Department of Biology and Biochemistry, University of Bath, Claverton Down, United Kingdom, **7** Centre hospitalier universitaire de La Réunion, Saint Denis, France

\* rophil@univ-reunion.fr (PR); emmanuel.bourdon@univ-reunion.fr (EB)

**Data Availability Statement:** All relevant data are within the manuscript and its Supporting Information files.

**Funding:** This work was supported by the Ministère de l'Enseignement Supérieur et de la

## Abstract

Diabetes is associated with a dramatic mortality rate due to its vascular complications. Chronic hyperglycemia in diabetes leads to enhanced glycation of erythrocytes and oxidative stress. Even though erythrocytes play a determining role in vascular complications, very little is known about how erythrocyte structure and functionality can be affected by glycation. Our objective was to decipher the impact of glycation on erythrocyte structure, oxidative stress parameters and capacity to interact with cultured human endothelial cells. *In vitro* glycated erythrocytes were prepared following incubation in the presence of different concentrations of glucose. To get insight into the *in vivo* relevance of our results, we compared these data to those obtained using red blood cells purified from diabetics or non-diabetics. We measured erythrocyte deformability, susceptibility to hemolysis, reactive oxygen species production and oxidative damage accumulation. Altered structures, redox status and oxidative modifications were increased in glycated erythrocytes. These modifications were associated with reduced antioxidant defence mediated by enzymatic activity. Enhanced erythrocyte phagocytosis by endothelial cells was observed when cultured with glycated erythrocytes, which was associated with increased levels of phosphatidylserine—likely as a result of an eryptosis phenomenon triggered by the hyperglycemic treatment. Most types of oxidative damage identified in *in vitro* glycated erythrocytes were also observed in red blood cells isolated from diabetics. These results bring new insights into the impact of glycation on erythrocyte structure, oxidative damage and their capacity to interact with endothelial cells, with a possible relevance to diabetes.

Recherche, the Université de La Réunion, the "Structure fédérative de recherche biosécurité en milieu tropical (BIOST) and by the European Regional Development Funds RE0001897 (EU-Région Réunion -French State national counterpart)."

**Competing interests:** The authors have declared that no competing interests exist.

**Abbreviations:** AGEs, advanced glycation (or glycoxidation) end products; ANOVA, analysis of variance; AU, arbitrary unit; DCFH-DA, dichlorofluorescein diacetate; ECL, enhanced luminol chemiluminescence; FACS, fluorescence-activated cell sorting; Flu-BA, Fluorescein boronic acids; $G_X$, erythrocytes incubated with × mM glucose; $G_0$, incubated in the absence of glucose; 4-HNE, 4-hydroxynonenal; HT50, 50% hemolysis time; MALDI, matrix-assisted laser desorption ionization; PBS, phosphate-buffered saline; ROS, reactive oxygen species.

## Introduction

Currently, more than 380 million people worldwide suffer from diabetes and this number is expected to double by 2035 [1]. Diabetes significantly enhances the risk of developing cardiovascular disease, which remains the leading cause of mortality in western countries [2]. Diabetes mellitus is responsible for the appearance of several microvascular and macrovascular complications such as coronary heart disease and ischemic strokes. Diabetic patients also exhibit a two- to three-fold increase in the risk of heart attacks and strokes [3]. Oxidative stress and oxidative modifications of proteins represent deleterious phenomena that have been implicated in the promotion of diabetic complications [2]. Oxidative stress was defined as an imbalance between oxidants such as reactive oxygen species (ROS) and antioxidants in favour of the oxidants, leading to a disturbance of redox signalling and molecular damage [4]. Chronic hyperglycemia in diabetes pathology leads to enhanced oxidative stress and damage to proteins such as glycation. This phenomenon is linked to the non-enzymatic attachment of a glucose molecule or derivatives to a free primary amine residue. Amadori rearrangement of the glycated protein gives rise to a heterogeneous class of deleterious compounds termed advanced glycation end-products (AGE) [5]. Glycation processes are enhanced in diabetics and affect mainly long half-life circulating proteins in particular hemoglobin [5]. Glycated hemoglobin (HbA1c) analysis is a clinical test routinely used to determine blood glucose exposure over a long period (weeks/months) in diabetics. Circulating glycated proteins exhibit altered structure and function and may play a pivotal and causative role in diabetes-associated vascular complications [6,7,8].

Human erythrocytes represent the most abundant and one of the most specialized cells in the body and their unique structural feature is constituted by the absence of nuclei, mitochondria and ribosomes [9]. The main function of erythrocytes is to transport of oxygen ($O_2$) through the human circulatory system [10]. Their role in oxygen transport and the presence of heme iron result in the formation of high levels of oxidizing radicals in erythrocytes [10]. To avoid oxidative stress, oxidizing radicals can be "detoxified" by antioxidant enzymes such as superoxide dismutase, catalase and glutathione peroxidase, which are commonly found in erythrocytes [10]. When oxidative stress occurs, oxidised proteins may be degraded by the 20S proteasome system, which was only recently described in erythrocytes [11,12]. Erythrocytes play an active role in the development of chronic vascular diseases [13]. They constitute the main solid particles present in blood that can 'squeeze' through narrow vessels thanks to the high deformability of their membrane. Erythrocytes are therefore the main blood component in contact with endothelial cells. Very recently, a direct relationship has been established between the erythrocyte width and coronary artery disease rate [14]. During atherosclerosis, a common complication in diabetic patients, erythrocytes can reach the atherosclerotic plaque after healed ruptures and thrombus formation. Rupture of micro vessels causing intraplaque hemorrhages can also bring erythrocytes into the plaque [15]. Very recently, high erythrocyte mortality levels (eryptosis) associated with enhanced phagocytosis by smooth muscle cells were found to be a promoting factor of oxidative stress in early-stage atheroma in people [16].

Despite the well-established implications of oxidative damage in diabetes disorder development and the active role of erythrocytes in vascular complication, very little is known about the impact of glycation on the structure of erythrocytes, their redox status and capacity to be phagocytosed by endothelial cells. Here, we hypothesized that glycation impairs erythrocyte structure, redox status, hemolysis sensibility and enhances its phagocytosis by cultured human endothelial cells. In light of the results presented in this paper we believe that enhanced glycation-mediated modification of erythrocytes and endocytosis by endothelial cells could play an important role in the development of the diabetes-linked vascular complications.

## Materials and methods

### Erythrocyte preparations

Experiments involving human erythrocytes were approved by our institutional review board at Centre Hospitalier Universitaire (La Réunion, France). Samples were obtained from the Alb-Ox ERMIES an ancillary, pilot study of ERMIES (NCT01425866). All data were analyzed anonymously.

Blood from healthy volunteers with informed consent was collected in EDTA tubes (BD vacutainer®). After centrifugation, erythrocytes were washed 3 times with a sterile isotonic solution (NaCl 0.15 M, pH 7) and suspended to 20% hematocrit in sterile PBS. This solution was subsequently diluted with glucose stock solutions made in PBS to form incubation mixtures of erythrocytes with 0, 5, 25 and 137 mmol/l glucose (corresponding to 0, 0.9, 4.5 and 25 mg/mL glucose, respectively). After 5 days of incubation at 37˚C, erythrocytes were washed 3 to 4 times with 0.15 M NaCl. For specific studies, erythrocytes from 12 type-2 diabetic individuals (HbA1c (%) = 11 ± 2.6) and from 9 non-diabetic individuals of the same age (HbA1c (%) = 4.4 ± 1) were obtained from the Alb-Ox ERMIES an ancillary, pilot study of ERMIES (NCT01425866). Erythrocytes were isolated as previously described and then suspended to 10% hematocrit in sterile PBS before incubation at 37˚C for 5 days. Erythrocyte preparations were either directly analysed by FACS and ektacytometry, lysed with an equivalent volume of distilled water (enzymatic activities, redox status) or subjected to "ghost" preparation (membrane oxidative damage). In lysates, optical density was measured at 280 nm and protein concentration was calculated according to a standard curve of hemoglobin (50–500 μg) and expressed in μg per μL. In membrane preparation, protein concentrations were measured using Bradford assay.

### HbA1c (%) level

Measurements were performed on our diluted erythrocyte preparations by using a high-pressure liquid chromatography method performed on an automated Hemoglobin analyser (D-10, Biorad) at the hospital of Saint Paul (La Réunion, France).

### Mass spectroscopy for average molecular mass determination

Glycation of both α and β hemoglobin subunits was analysed by matrix-assisted desorption/ionization time-of-flight mass spectrometry (MALDI-TOF MS) for mass shift determination as previously described [17]. Mass spectra were obtained in three independent experiments. On each spectrum, the four main peaks for α and β haemoglobin subunits and their glycated forms were identified. For each peak, the mass (m/z) and Δmass between non-glycated and glycated subunits were obtained. Relative intensity of each peak was calculated as follow: % glycation = (intensity glycated-hemoglobin / intensity hemoglobin) x 100.

### 5-hydroxymethylfurfural level

The thiobarbituric acid (TBA) assay was used for 5-hydroxymethylfurfural (5-HMF) quantification in our different preparations according to the protocol detailed in Murtiashaw et al. [18]. Briefly, 1 ml of diluted erythrocytes was hydrolysed at 100˚C for 1 h after addition of 0.5 ml of 0.3 N oxalic acid. After cooling to RT, 0.5 ml of 40% trichloroacetic acid was added and thoroughly mixed before centrifugation for 15 min at 3000 g. Then, 0.5 ml of 0.05 M TBA was added to 0.4 ml of supernatant and the mixture was incubated at 40˚C for 30 min. The absorbance was read at 443 nm. The 5-HMF concentration was determined by using its molar extinction coefficient ($4.10^8$) and expressed as mmol of 5-HMF per milligram of protein.

## Early glycation product determination

Fluorescein boronic acids (Flu-BA) were used to detect early glycation products (EGP) in our erythrocyte preparations. Flu-BA was prepared following the protocols detailed in Pereira Morais et al. [19]. Boronic acids specifically interact with fructosamine-modified proteins via interaction with the cis-1,2-diol containing adducts [19]. This probe was recently used to quantify carbohydrate modifications in tissue extracts [20,21].

Prior to flow cytometry analysis, erythrocyte preparations (approximately $10^6$ cells) were incubated with 25 μM of Flu-BA or fluorescein in binding buffer (BioLegend) for 30 min at RT. After incubation, cells were pelleted by centrifugation (1000 rpm, 5 min) and supernatant was discarded, and labelled cells were resuspended in PBS. Fluorescence was measured by FACS (Beckman Coulter's CytoFLEX and Cytexpert software) with an excitation and emission wavelengths of 488 nm and 530 nm, respectively. The level of early glycation products were determined after mean fluorescence normalization of erythrocytes probed with fluorescein only and expressed as a percentage relative to the control (non glycated erythrocytes G0).

## Free radical-induced hemolysis test

The capacity of erythrocyte preparations to resist lysis induced by an oxidative stress was investigated by using the *in vitro* free radical-induced blood hemolysis assay. Hemolysis was induced using a water-soluble free radical generator, 2,2'-azo-bis 2-aminodinopropane (AAPH, Sigma). 135 μL of diluted erythrocyte preparations (approximately $10^8$ erythrocytes) were added to each well of a 96-well plate. Hemolysis was started by adding 40μL of 0.5 M AAPH to each well and the turbidimetry at 450 nm was recorded every 10 minutes using a temperature controlled microplate reader at 37˚C (Fluostar, BMG Labtech). For each condition, the time to 50% of maximal hemolysis (HT50) was determined in triplicate. In other work, the measurement of HT50 was shown to be very reproducible: 1.32% and 3.85% intra- and inter-assay coefficients, respectively [22].

## Enzymatic activities

SOD activity was measured by monitoring the rate of acetylated cytochrome c reduction by superoxide radicals generated by the xanthine/xanthine oxidase system as published in [23]. Measurements were performed using the reagent buffer (xanthine oxidase, xanthine (0.5 mM), cytochrome c (0.2 mM), $KH_2PO_4$ (50 mM), EDTA (2 mM), pH 7.8) at 25˚C. The kinetics of cytochrome c reduction were monitored by spectrophotometry at 560 nm. SOD activities calibrated relative to a standard curve of SOD up to 6 unit/mg.

The catalase activity assay was carried on 40 μg of protein lysate in 25 mM Tris–HCl (pH 7.5), as recently described [23]. Blanks were measured at 240 nm just before adding 80 μL of $H_2O_2$ (10 mM final) to start the reaction. The kinetics of $H_2O_2$ reduction were monitored by measuring the absorbance every 5 s at 240 nm for 1 min and catalase activity was calibrated relative to a standard curve of increasing amount of catalase between 12.5 and 125 units/ml. Catalase activity was expressed as international catalytic units per mg of protein.

Chymotrypsin-like activity of the proteasome was assayed using the fluorogenic peptide (Sigma-Aldrich, St Louis): Suc-Leu-Leu-Val-Tyr-7-amido-4-methylcoumarin (LLVYMCA at 25 mM), as described previously [24].

Peroxidase activities of cell lysates were assessed according to the method of Everse *et al* [25]. A reaction mixture was prepared with 50 mM citrate buffer, 0.2% o-dianisidine and samples diluted 1/100. The reaction was initiated by adding 20 mM $H_2O_2$. Peroxidase activity was determined by measuring the absorbance at 450 nm at 25˚C for 3 min. Peroxidase activity was expressed as international catalytic units per mg of protein.

## Dot-blots

Four microliters of erythrocyte lysate (approximately 20 μg of proteins) were spotted onto a dry nitrocellulose membrane. The membrane was air-dried for 5 minutes and total protein was stained using Ponceau red dye. The membrane was initially blocked with PBS/Tween 20 0.1% (v/v)/ milk 5% for 3 hours at room temperature and then sequentially probed for another 3 hours with a primary antibody directed against 4-HNE (1:1000; ab46545; Abcam, Cambridge MA) or rabbit anti AGE antibody (Abcam, Ab23722). This was followed by secondary antibody incubation for at least 1 hour (1:2000; Peroxidase AffiniPure Goat Anti-Rabbit IgG (H +L); Jackson Immunoresearch Laboratories Inc; 111-035-003). Between each step, membranes were washed three times with PBS/Tween 20 0.1% (v/v). Detection was performed using the enhanced chemiluminescence reagent (ECL®, GE Healthcare). Signal intensities were quantified using the freeware ImageJ (version 1.32j) available from the internet website: http://rsb.info.nih.gov/ij/.

## Endothelial cell culture and stimulation

The human endothelial EA.hy926 cell line was obtained from the American tissue culture collection (CRL-2922) and was cultured in DMEM supplemented with 10% Fetal Bovine Serum (FBS), penicillin (100 units/ml), streptomycin (100 μg/ml), L-glutamine (2 mM) and HAT (hypoxanthine 100 μmol/L; aminopterin 0.4 μmol/L and thymidine 16 μmol/L). Cells were grown in a 5% $CO_2$ incubator at 37°C in a humidified atmosphere. Approximately 100 000 cells were plated in 24 cell plates. When cells reached confluency, they were treated in the absence (control PBS) or presence of 71 μl/cm² of the different erythrocytes preparations for 24 h. Internalized erythrocytes were detected using the 2,7-diaminofluorene (DAF) reagent which quantifies the pseudo-peroxidase activity of RBC hemoglobin. After 24-hours of incubation with erythrocytes, endothelial cells were washed 3 times with PBS and treated with water for 3 minutes to induce lysis (by hypotonic shock) of any fixed erythrocytes at the cellular membrane surface and supernatant was discarded.

Endothelial cells were then lysed with PBS/Triton X100 for 3 minutes to release the cytosolic fraction containing internalized erythrocytes and their hemoglobin. DAF solution was prepared extemporaneously by dissolving 10 mg of DAF reagent in 10 ml of tris HCl 0.2 N with 9% acetic acid supplemented with 20 μl of 30% hydrogen peroxide just before use. 100 μl of DAF solution was added to 40 μl of cytosolic samples previously transferred to a 96-well plate. Optical density (OD) was measured at 620 nm. Internalized erythrocytes were quantified relative to calibration erythrocyte standards (27–5760 cells/μl) and results were expressed as erythrocyte number.

## Flow cytometry assays

Erythrocyte shape, eryptosis evaluation and intracellular reactive oxygen species (ROS) production in our different erythrocyte preparations were measured by flow cytometry using Beckman Coulter's CytoFLEX and Cytexpert software. A specific erythrocyte cell population was selected by gating and could be characterized by its typical location in a forward scatter (FSC) *versus* a side scatter (SSC) parameter graph. For phosphatidylserine exposure determination, erythrocytes were incubated with 2 μg/ml Annexin V-FITC in binding buffer (BioLegend) for 30 min at RT before flow cytometry analysis. Annexin V protein exhibits a high affinity for phosphatidylserine (PS) and was measured with an excitation wavelength of 488 nm and an emission wavelength of 530 nm. For evaluation of intracellular reactive oxygen species (ROS) production, erythrocytes were incubated with 2 μM of the fluorescent probe

dihydroethidium (DHE; Sigma-Aldrich, D7008) or dichlorodihydrofluoresceindiacetate (DCFH-DA; Sigma-Aldrich, D6883) for 30 min at RT.

### Ektacytometry

The determination of erythrocytes membrane deformability was performed using an ektacytometer (LORCCA MaxSis, Mechatronics, The Netherlands) which measures the elongation of red blood cells at increasing shear stress. Red blood cells suspended at 10% hematocrit were diluted 200 times in an iso-osmolar solution of polyvinylpyrrolidone buffer (PVP, viscosity 28.6 mPa/s). Deformation was expressed as an elongation index (EI) was calculated for 19 shear-stresses between 0.30 and 80 Pa (increasing rotation speed) as follows: EI = (A-B) / (A+B), where A and B represent the length and the width of the ellipsoid diffraction pattern, respectively. The deformability curve was obtained by plotting the calculated values for EI versus the shear stress [26].

### Statistical analysis

Data are expressed as the mean ± standard deviation (SD) or as the mean ± standard error of the mean (SEM) from at least three independent experiments performed in triplicate. Statistical analyses were performed with Prism (GraphPad Software Inc., San Diego, CA, USA). Statistical significance was determined using the Student's t-test or one-way ANOVA followed by Dunnett's test, with a p-value < 0.05 required for significance.

## Results

### Impact of in vitro glycation on erythrocyte morphology and redox status

Experiments were designed to determine whether short-term incubation of erythrocytes with increasing concentrations of glucose may affect their morphology and redox status.

To characterize the glycation level of our erythrocyte preparations, the percentage of HbA1c and the 5-HMF concentration were measured in the lysates of our different preparations (Table 1). A significant increase in HbA1c percentage was observed in erythrocytes incubated with 137 mM of glucose (p<0.05) compared to erythrocytes incubated in the absence of glucose (G0). Interestingly, the value obtained for G137 erythrocytes (7%) is highly similar to HbA1c values observed in diabetic patients. Indeed, 6.5% in HbA1c corresponds to the threshold used to diagnose persons with diabetes [27]. Similar results were observed with the 5-HMF concentrations in the different glycated erythrocyte preparations (Table 1). This intermediate formed from carbohydrates such as glucose is a good indicator of protein glycation [28].

Early glycation product (EGP) accumulation in erythrocyte preparations was evaluated using a specific fluorescent probe (Flu-BA) developed by our group [19].

**Table 1.** *In vitro* **incubation with high glucose concentration significantly enhances erythrocyte glycation.** The impact of erythrocyte incubation in the presence of enhanced glucose concentration on the percentage of glycated hemoglobin (% HbA1c), the 5-hydroxymethylfurfural and the early glycation product (EGP) levels were determined as described in material and method section. GX corresponds to erythrocytes incubated with X mM glucose and G0 corresponds to erythrocytes incubated in the absence of glucose. Data are expressed as mean ± SEM (n = 6 to 8 independent replicates) and statistical analyses were performed using One-way ANOVA followed by Dunnett's test. *p<0.05, **p<0.01 in comparison with G0.

| | G0 | | | G5 | | | G25 | | | G137 | | |
|---|---|---|---|---|---|---|---|---|---|---|---|---|
| **HbA1c (%)** | 5.22 | ± | 0.4 | 5.18 | ± | 0.4 | 4.9 | ± | 0.47 | 7.06 | ± | 1.65* |
| **5-HMF (% G0)** | 100 | ± | 14.4 | 97.2 | ± | 10.1 | 102 | ± | 14.8 | 155 | ± | 31.5** |
| **EGP (% G0)** | 100 | ± | 19.3 | 129.8 | ± | 14.4 | 135.3 | ± | 20.4 | 457 | ± | 63.2** |

EGP levels determined after fluorescence normalization of erythrocytes probed with fluorescein only are reported in Table 1. A significant increase in EGP accumulation was observed in erythrocytes incubated with 137 mM of glucose ($p<0.01$) compared to erythrocytes incubated at G0.

Flow cytometry of fluorescein boronic acid-labelled erythrocytes detected a distinct subpopulation of cells that appeared after treatment with high concentrations of glucose (Fig 1).

To further characterize the glycation level in our erythrocyte preparations, the proportion of glycated forms of both α and β hemoglobin was determined by mass spectrometry. Representative figures of the mass spectra obtained for α- and β -hemoglobin subunits and their glycated forms are presented in the supplementary materials (cf S1 Fig). The relative intensity of

**1**

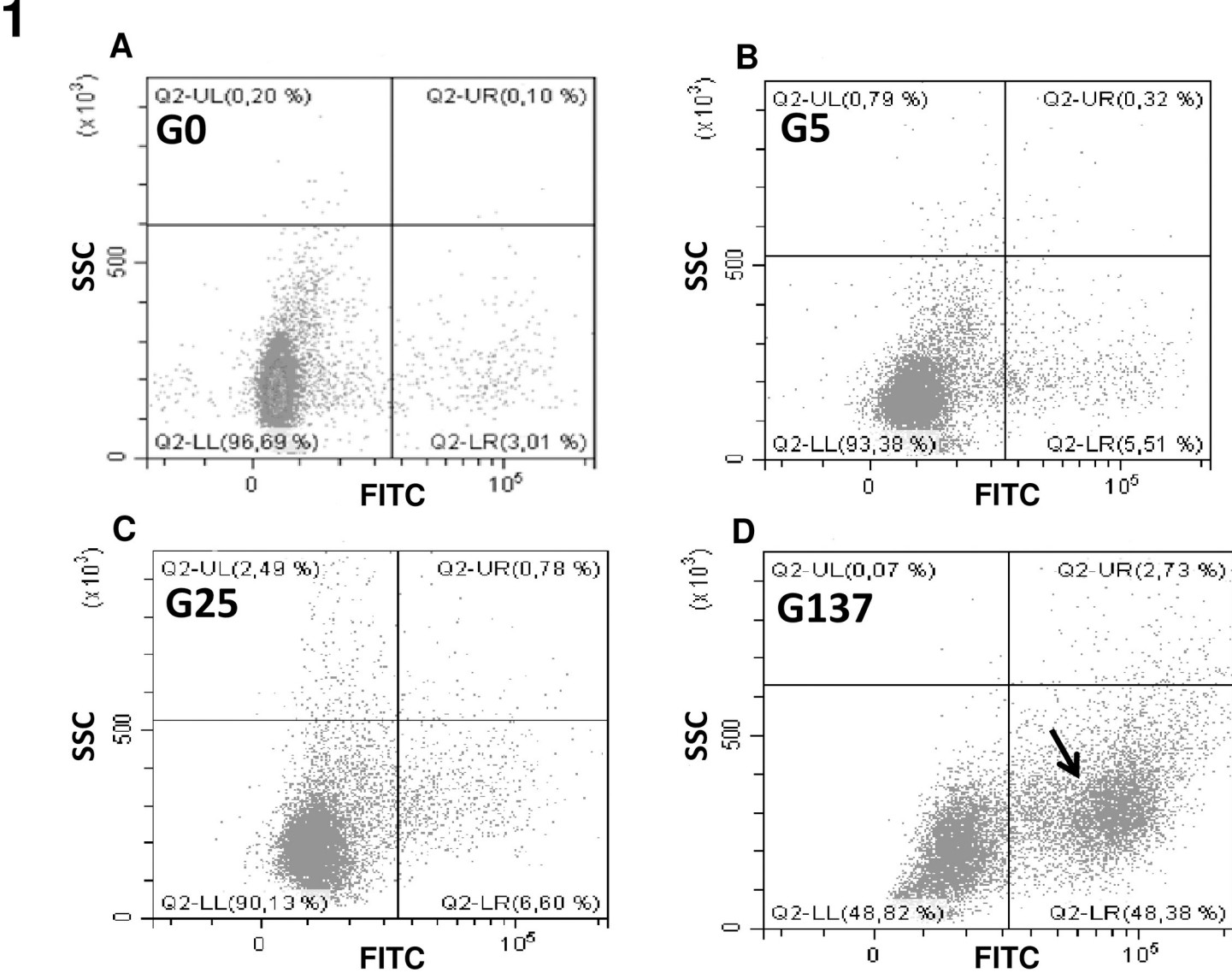

**Fig 1. Early glycation product detection by using fluorescent boronic acids.** Cytometry analysis of our erythrocyte preparations was performed as described in material and method section. Following probing with fluorescent boronic acids, erythrocyte populations were gated according to cell location in a side scatter (SSC) parameter *vs*. FITC fluorescence. Black arrow evidences the specific population of glycated positive erythrocytes that become predominant when they were incubated with increasing concentrations of glucose.

m/z peaks corresponding to glycated and non-glycated forms for both subunits, and the corresponding percentage of glycation are presented in Table 2. Our results show erythrocytes incubated with 137 mM of glucose are significantly more glycated in both hemoglobin subunits (α and β) than G0 erythrocytes. It is worth noting that for all erythrocyte preparations, the glycated forms of the Hb subunits display a ~200 Da increase in mass. The increase in glycation was also confirmed by significant increases in AGE content in dot blots (Fig 2A and 2B) and western blots (Fig 2C and 2D).

The capacity of erythrocytes to resist oxidative stress was examined using *in vitro* free radical-induced blood hemolysis. Results were expressed as the half time of hemolysis (HT50) following our erythrocyte preparations "attack" with a free radical generator (AAPH). Erythrocytes incubated with 5 or 25 mM of glucose (G5, G25) exhibited a significantly higher HT50 compared to G0 (Fig 2E). This was suggestive of a protective membrane-stabilising effect of low concentrations of glucose which, interestingly, was no longer observed when incubated at 137 mM glucose (Fig 2E).

Due to their biconcave shape, erythrocytes are able to deform and pass through small vessels like capillaries [9]. The deformability of the different erythrocyte preparations was analysed using the ektacytometry method and results are displayed in Fig 2F and 2G. Changes in the elongation index revealed a loss of cellular deformability in response to increasing shear stress when erythrocytes were incubated with increasing concentrations of glucose. A significant decrease of deformability was observed in glycated erythrocytes compared to erythrocytes incubated in the absence (G0) or presence of low concentration of glucose (G5). These data clearly indicate that the glycation phenomenon induced by the hyperglycemic incubations renders erythrocytes less deformable (about four times less) and more rigid than erythrocytes incubated under low glycemic conditions. It is worth noting that impaired deformability of glycated erythrocytes was also observed under low shear stress conditions (3 Pa), which are similar to those that can be encountered under standard physiological conditions [29].

Erythrocyte morphology and deformability were subsequently investigated in order to get further insights into erythrocyte fragility. Flow cytometry was used to determine the relative size of our different erythrocyte preparations. By using the FSC and SSC parameters, each

**Table 2. Characterisation of glycation percentage in the different erythrocyte preparations by mass spectrometry.** Data for the four main peak of the mass spectra corresponding to α-hemoglobin (α-Hb; 15130 Da), glycated α-hemoglobin (g-α-Hb; 15330 Da), β-hemoglobin (β-Hb; 15890 Da), glycated β-hemoglobin (g-β-Hb; 16100 Da) (S1 Fig—see supplemental data). G0, G5, G25 and G137 represent the four conditions of incubation to which erythrocytes were subjected: 0, 5, 25 and 137 mmol/l glucose, respectively. Δmass and % glycation were calculated as explained in the methods section. Results are the mean ± SD and statistical analyses were performed using Tukey's post hoc analysis following a significant one way ANOVA: $^{***}p<0.001$, $^{**}p<0.01$, $^{*}p<0.05$ (vs. G0), $^{###}p<0.001$, $^{#}p<0.05$ (vs. G5).

| | α-hemoglobin | | | | | |
|---|---|---|---|---|---|---|
| | α-Hb | | glycated-α-Hb | | | |
| | mass/z | Intensity | mass/z | Intensity | Δ mass | % glycation |
| G0 | 15132.2 ± 3.2 | 98.9 ± 2.9 | 15334.1 ± 3.4 | 46.7 ± 3.8 | 201.8 ± 1.2 | 48.4 ± 1.2 |
| G5 | 15132.4 ± 3.8 | 100.0 ± 0.1 | 15334.8 ± 2.9 | 48.7 ± 2.9 | 202.4 ± 1.2 | 48.6 ± 0.7 |
| G25 | 15134.9 ± 2.6 | 98.3 ± 5.1 | 15335.5 ± 2.5 | 50.3 ± 4.0 | 200.6 ± 0.9 | 51.2 ± 0.7 |
| G137 | 15139.2 ± 3.2 | 95.2 ± 7.9 | 15334.0 ± 3.8 | 54.9 ± 6.4 | 194.8 ± 1.9 | 57.8 ± 1.3$^{**#}$ |
| | β-hemoglobin | | | | | |
| | β-Hb | | glycated-β-Hb | | | |
| | mass/z | Intensity | mass/z | Intensity | Δ mass | % glycation |
| G0 | 15891.5 ± 3.7 | 77.5 ± 16.4 | 16093.1 ± 11.9 | 50.2 ± 8.2 | 201.8 ± 11.5 | 66.6 ± 1.9 |
| G5 | 15892.5 ± 4.2 | 78.6 ± 10.8 | 16091.7 ± 7.4 | 52.6 ± 8.4 | 201.8 ± 11.5 | 67.0 ± 1.3 |
| G25 | 15894.5 ± 3.6 | 78.3 ± 15.4 | 16089.0 ± 3.7 | 54.5 ± 7.8 | 194.5 ± 4.2 | 70.0 ± 1.4$^{*}$ |
| G137 | 15897.2 ± 3.6 | 89.9 ± 12.1 | 16088.5 ± 3.9 | 67.1 ± 11 | 191.3 ± 3.8 | 74.3 ± 1.2$^{***####}$ |

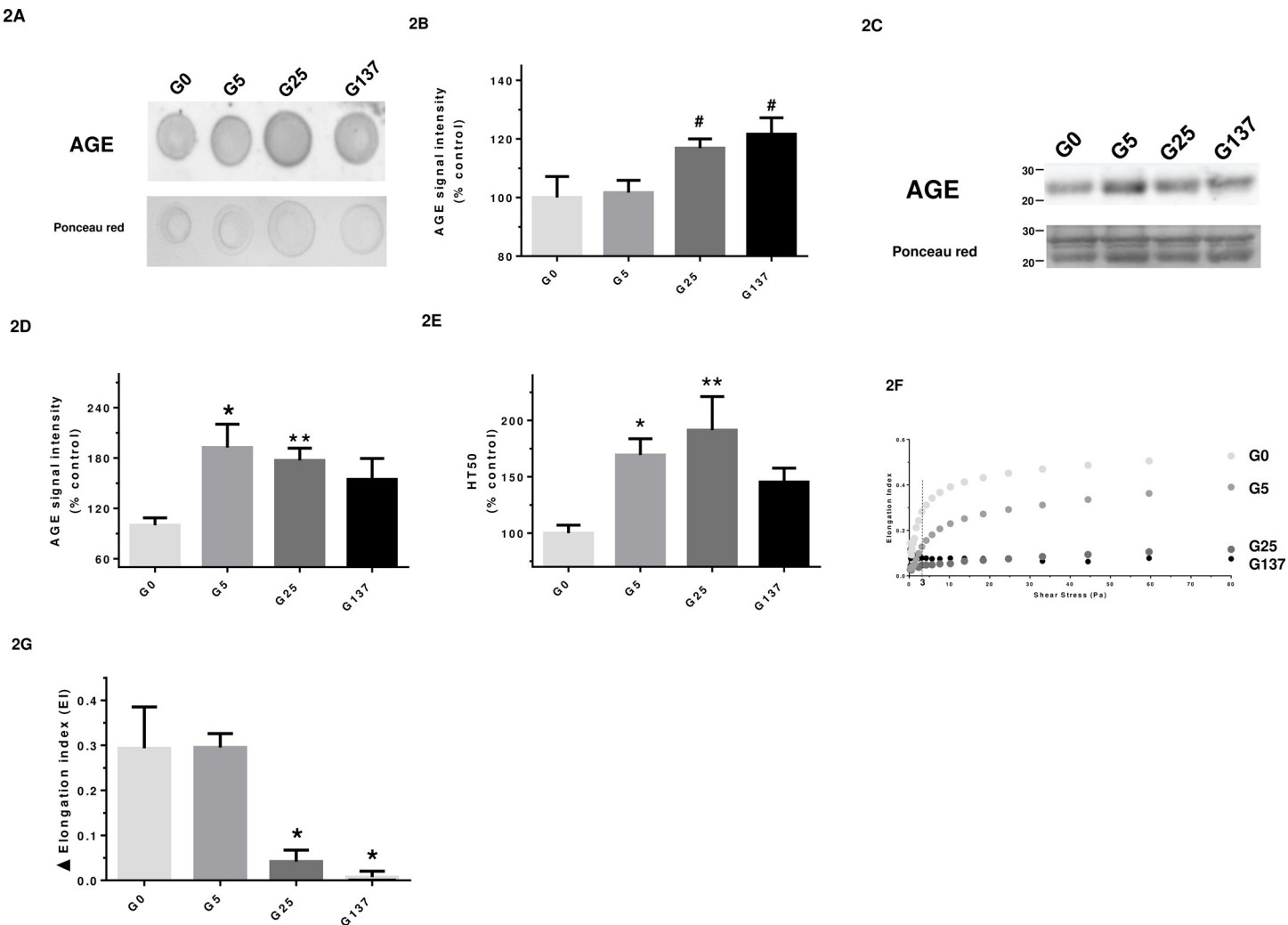

**Fig 2. Glycated erythrocytes exhibit an impaired deformability capacity.** In this figure, G0, G5, G25 and G137 represent the four conditions of incubation to which erythrocytes were subjected: 0, 5, 25 and 137 mmol/l glucose, respectively. (A) Representative AGE dot blot performed on lysate preparations (n = 4); (B) Quantification of AGE signal normalized with Ponceau S signal in the different erythrocyte preparations. Results are expressed as mean ± SEM of 3 to 4 experiments performed independently. #p<0.05 *vs*. G5 (Student's t test, n = 3 to 4); (C) Representative AGE western blot performed on lysate preparations (n = 4); (D) Quantification of AGE signal normalized with ponceau red signal in the different erythrocyte preparations. Results are expressed as mean ± SEM of 4 experiments performed independently. *p<0.05, **p<0.01 vs. G0 (Student's t test, n = 4). (E) HT50 was measured by the free-radical hemolysis test as described in method section. Results are expressed as mean ± SEM of 5 to 8 experiments performed independently. *p<0.05, **p< 0.01 indicates a significant difference *vs*. G0 (One-way ANOVA followed by Dunnett's test) n = 5 independent analyses; (F) Curves correspond to the elongation index of erythrocytes determined by LORRCA measurement as a function of shear stress intensity (Pa); (G) Histograms correspond to the calculated variation in elongation index (delta EI) reflecting capacity of erythrocytes to deform when submitted to a shear stress ranking from 0 to 80 Pa. Results are expressed as mean ± SEM. *p<0.05 indicates a significant difference as compared to G0 (One-way ANOVA followed by Dunnett's test) n = 3 independent replicates.

preparation containing a mix of erythrocytes was analysed and specific populations were gated according to cell size and granularity. We observed the formation of an additional erythrocyte population (highlighted by an arrow in Fig 3A), which became dominant when incubated at higher concentrations of glucose. This potentially represents a population of erythrocytes engaged in an accelerated aging process induced by the hyperglycemic treatment [30].

Hyperglycemic conditions are known to be associated with increased ROS production (5). Therefore, we evaluated the impact of *in vitro* glycation on erythrocyte redox status. The impact of *in vitro* glycation on erythrocyte redox balance was first determined by the analysis

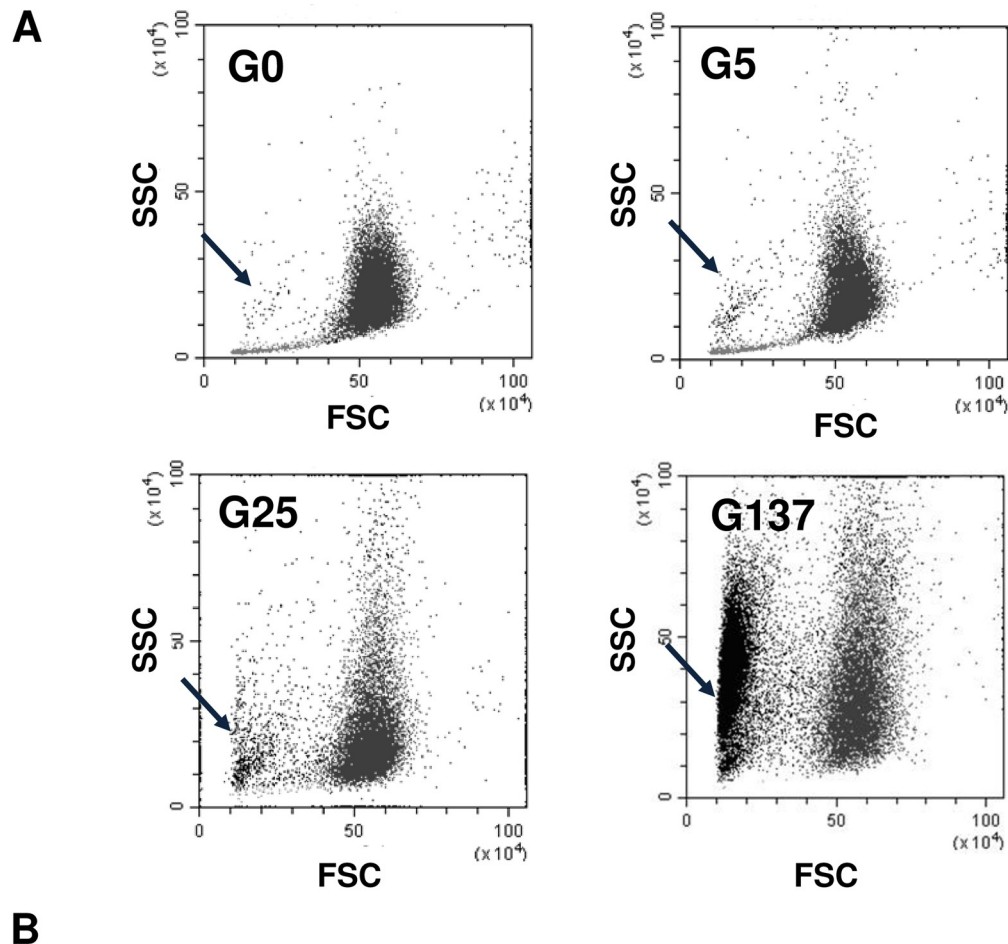

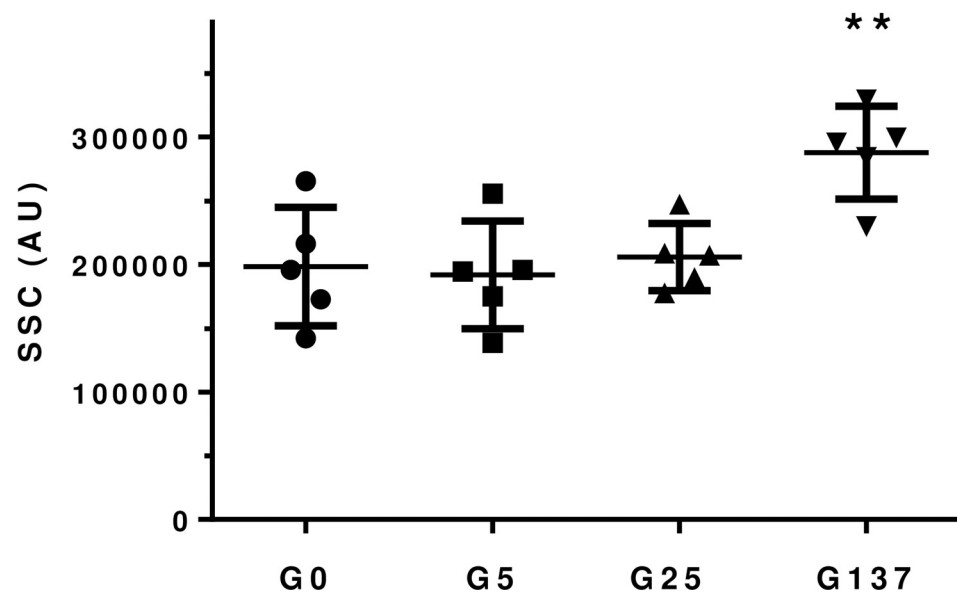

**Fig 3. Glycation alters erythrocyte morphology.** Cytometry analysis of our erythrocytes preparation was performed as described in material and method section. GX corresponds to erythrocytes incubated with X mM glucose and G0 corresponds to erythrocytes incubated in the absence of glucose. (A) Erythrocyte populations were gated according to cell location in a forward scatter (FSC) versus a side scatter (SSC) parameter. Black arrow evidences the specific population of glycated altered erythrocytes that become predominant when they were incubated with increasing concentrations of glucose. (B) SSC parameters of our erythrocytes preparation were performed as described in material and method section $^{**}$p<0,01 indicates a significant difference *vs*. G0 (Student's t test).

of intracellular ROS formation using a specific fluorescent probe in cell lysate. The dihydroethidium molecule (DHE) can penetrate erythrocytes and emit fluorescence when oxidised by free radicals. Increase in DHE fluorescence in erythrocytes was observed when incubated with increasing concentrations of glucose (Fig 4A). This enhanced intracellular free radical formation reached our significance threshold in erythrocytes glycated with 137 mM of glucose (p<0.01, *vs*. G0). The levels of the oxidative damage-indicating biomarker 4-HNE were significantly higher in erythrocytes that were glycated with 25 or 137 mM of glucose compared to those exposed to 5 mM of glucose (Fig 4B and 4C).

In order to gain further insight into the redox status of our erythrocyte preparations and the origin of enhanced ROS formation and oxidative damage in glycated erythrocytes, antioxidant enzyme and proteasome activities were measured (Table 3). Whilst SOD and catalase activities appeared not to be significantly different in our erythrocyte preparations, a significantly reduced peroxidase activity was measured in erythrocytes that were incubated with 137 mM of glucose (-33%, p<0.05 *vs*. G25). This reduced peroxidase activity could explain the enhanced ROS formation in G137 erythrocytes. Similarly, a significant reduction of the chymotrypsin-like activity of the proteasome was measured in G137 erythrocytes (-74%, p<0.05 *vs*. G5).

Enhanced intracellular oxidative stress could be a determining signal for glycated erythrocytes to initiate eryptosis, the programmed cell death for anuclear erythrocytes. To confirm this hypothesis, we investigated whether glycated erythrocyte preparations exhibit phosphatidylserine exposure, which is a measure of eryptosis [31]. Incubations with 25 mM and 137 mM of glucose lead to a significantly higher phosphatidylserine exposure, in a dose-dependent manner, when compared to erythrocytes incubated in the absence of glucose (Fig 4D). These results clearly indicate that eryptosis is enhanced in glycated erythrocytes. Phosphatidylserine exposure associated with membrane rigidity appears to be critical factors of red blood cell clearance. Effects of glycated erythrocytes were further investigated by analysing their capacity of being phagocytosed by the human endothelial EA.hy926 cell line.

Fig 4E shows that glycation of erythrocytes with high glucose concentration (25 mM and 137 mM) induced their phagocytosis by endothelial cells, as illustrated by high levels of internalized red blood cells. These results can be related to the enhanced eryptosis of red blood cells when glycated. Endothelial cells viability did not seem to be affected by the enhanced phagocytosis of glycated erythrocytes.

### *In vivo* modifications of erythrocytes in diabetic vs non-diabetic individuals

We compared results from our *in vitro* model of glycation, to those of erythrocytes isolated from diabetic or non-diabetic individuals in terms of both morphology and redox status.

Erythrocytes from diabetic patients were significantly less deformable than erythrocytes from non-diabetics (Fig 5A and 5B). Indeed, the variation of elongation index in response to an increasing shear stress appears to be lower for erythrocytes isolated from diabetic persons compared to non-diabetics. A significant higher SSC was measured for erythrocytes when

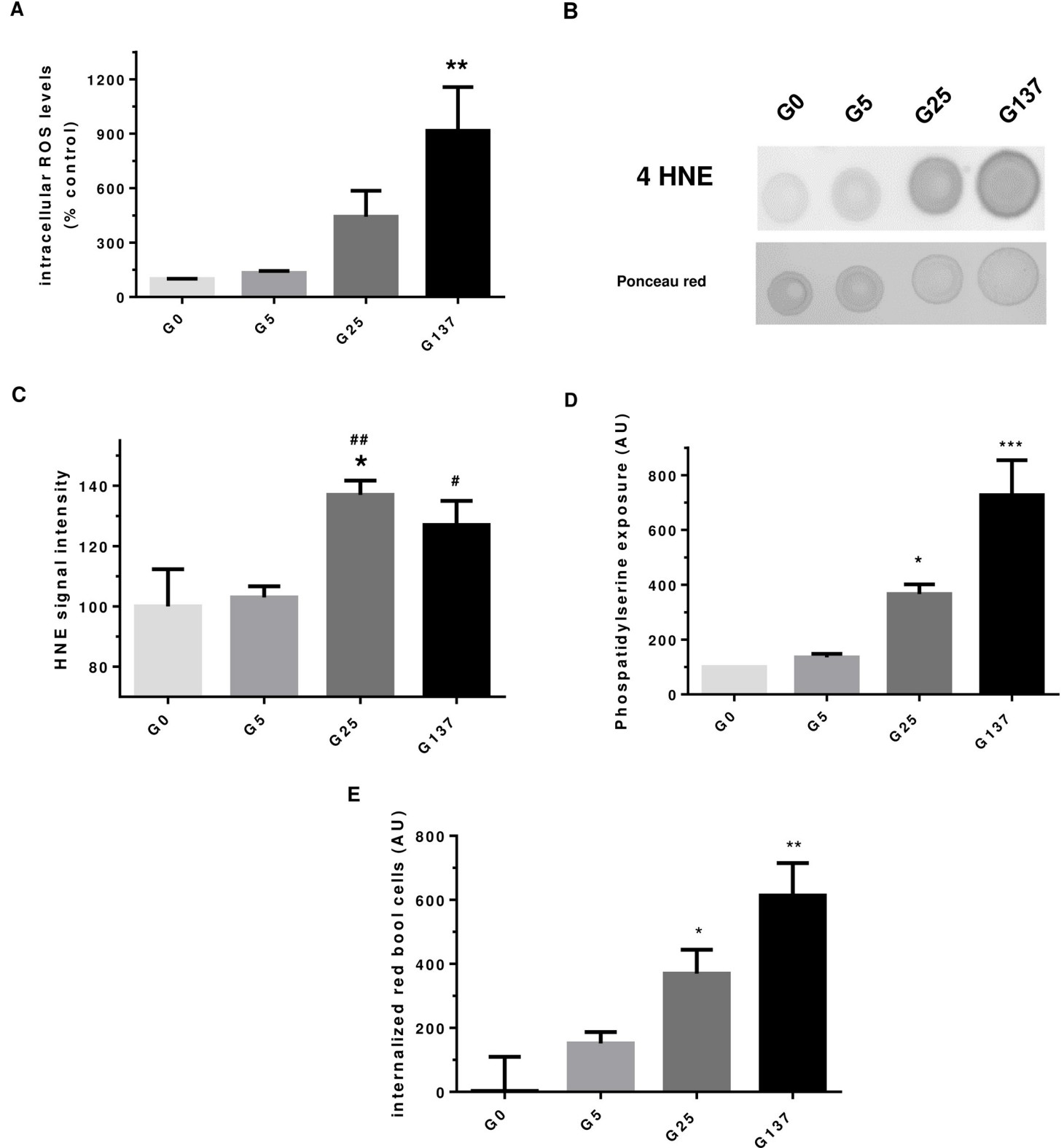

**Fig 4. Enhanced oxidative stress and damages in glycated erythrocytes.** In this figure, G0, G5, G25 and G137 represent the four conditions of incubation to which erythrocytes were subjected: 0, 5, 25 and 137 mmol/l glucose, respectively. (A) Intracellular ROS formation levels in erythrocyte preparation was determined using DHE probe by cytometry. Results are expressed as mean ± SEM (n = 4), **p<0.01 indicates a significant difference *vs.* G0 (one way ANOVA followed by Dunnet's test). (B) 4-HNE dot blot image is representative of four dot blot experiments. (C) 4-HNE signal quantification was expressed as mean ± SEM (n = 4), *p<0.05 (*vs.* G0), #p<0.05,

##p<0.01 (*vs*. G5) using Student's t test. (D) Phosphatidyl serine (PS) exposure in erythrocytes preparations was evaluated by cytometry as described in method section. Data are expressed as mean ± SEM, *p< 0.05, ***p< 0.001 *vs*. G0 (one-way ANOVA followed by Dunnett's test, n = 4). (E) Internalized red blood cell in cultured EA. hy926 cell lines was determined by DAF assay and are expressed in arbitrary unit as mean ± SEM (n = 3), *p < 0.05, **p< 0.01 (one-way ANOVA followed by Dunnett's test).

purified from diabetics (+38% ± 23.4, p<0.05 *vs*. ND), whereas the FSC value was not impacted (Figs 5C and S2). This result is in favour of an altered erythrocyte structure and shape when isolated from diabetics. With respect to the redox status (Fig 5D), a significant increase in intracellular ROS formation, probed by DHE, was evidenced in erythrocytes that were purified from diabetics (+39%, p<0.05 *vs*. ND). Similar and significant result was also evidenced by using DCFDA probe (Fig 5E).

Three oxidative parameters were determined in erythrocytes from diabetic and non-diabetic persons. No variation in 4-HNE levels was observed, while a significant increase in AGE formation was seen in erythrocytes from diabetics (Fig 5F and 5G). In addition, accumulation of advanced oxidation product (AOPP), was found to be higher (+18%) in erythrocytes from diabetics but this increase did not reach significance.

Although catalase and SOD activities appeared not to be significantly different between both groups (ND *vs*. D), a significant reduced peroxidase activity was measured in erythrocytes isolated from diabetic persons (Table 4). This reduction in peroxidase activity, associated with high glucose concentration, could explain the enhanced ROS formation in erythrocytes isolated from diabetic persons. Conversely, the chymotrypsin-like activity of the proteasome appeared to be significantly enhanced in erythrocytes purified from diabetics compared to those from non-diabetics (Table 4).

Proteasome activity may be activated under moderate oxidative stress [32]. *In vivo*, if oxidative stress is higher in erythrocytes from diabetics than from non-diabetics, no variation was measured in term of 4-HNE accumulation. Enhanced oxidative stress in *in vivo* glycated erythrocytes may be significant and moderate enough to trigger proteasome LLVY activation.

Finally, eryptosis in erythrocytes from diabetics or non-diabetics was investigated (Fig 5H). Erythrocytes from diabetic persons exhibit a tendency for a higher phosphatidylserine exposure than erythrocytes isolated from non-diabetic persons. This is in favour of a triggered eryptosis phenomenon in erythrocytes when *in vivo* glycated.

## Discussion

Despite the fact that erythrocytes represent a key player in vascular complications, very little is known about how structure, redox status and capacity of erythrocytes to be phagocytosed by endothelial cells can be affected by glycation. In this study we revealed that *in vitro* glycation

**Table 3. Effect of glycation on erythrocyte enzymatic activites involved in redox balance and in oxidised protein degradation.** Catalase, superoxide dismutase, peroxidase and chymotrypsine-like activity of the proteasome (LLVY). Enzymatic activities were determined as described in the material and methods section. G0, G5, G25 and G137 represent the four conditions of incubation to which erythrocytes were subjected: 0, 5, 25 and 137 mmol/l glucose, respectively. Results are expressed as mean ± SD (n = 4 to 9) and statistical analyses were performed using Tukey's post hoc analysis following a significant one way ANOVA: * effect of erythrocyte glycation (vs. G 5): * p<0.05. #p<0.05 (vs. G25).

| | G0 | | | G5 | | | G25 | | | G137 | | |
|---|---|---|---|---|---|---|---|---|---|---|---|---|
| **SOD** AU/mg prot | 100 | ± | 28 | 109 | ± | 42 | 121 | ± | 40 | 140 | ± | 73 |
| **Catalase** AU/mg prot | 100 | ± | 27 | 105 | ± | 30 | 146 | ± | 53 | 131 | ± | 82 |
| **Peroxidase** AU/µg prot | 100 | ± | 20 | 88 | ± | 31 | 106 | ± | 49 | 71 | ± | 26# |
| **LLVY** AU/µg prot | 100 | ± | 10 | 129 | ± | 73 | 110 | ± | 63 | 33 | ± | 37* |

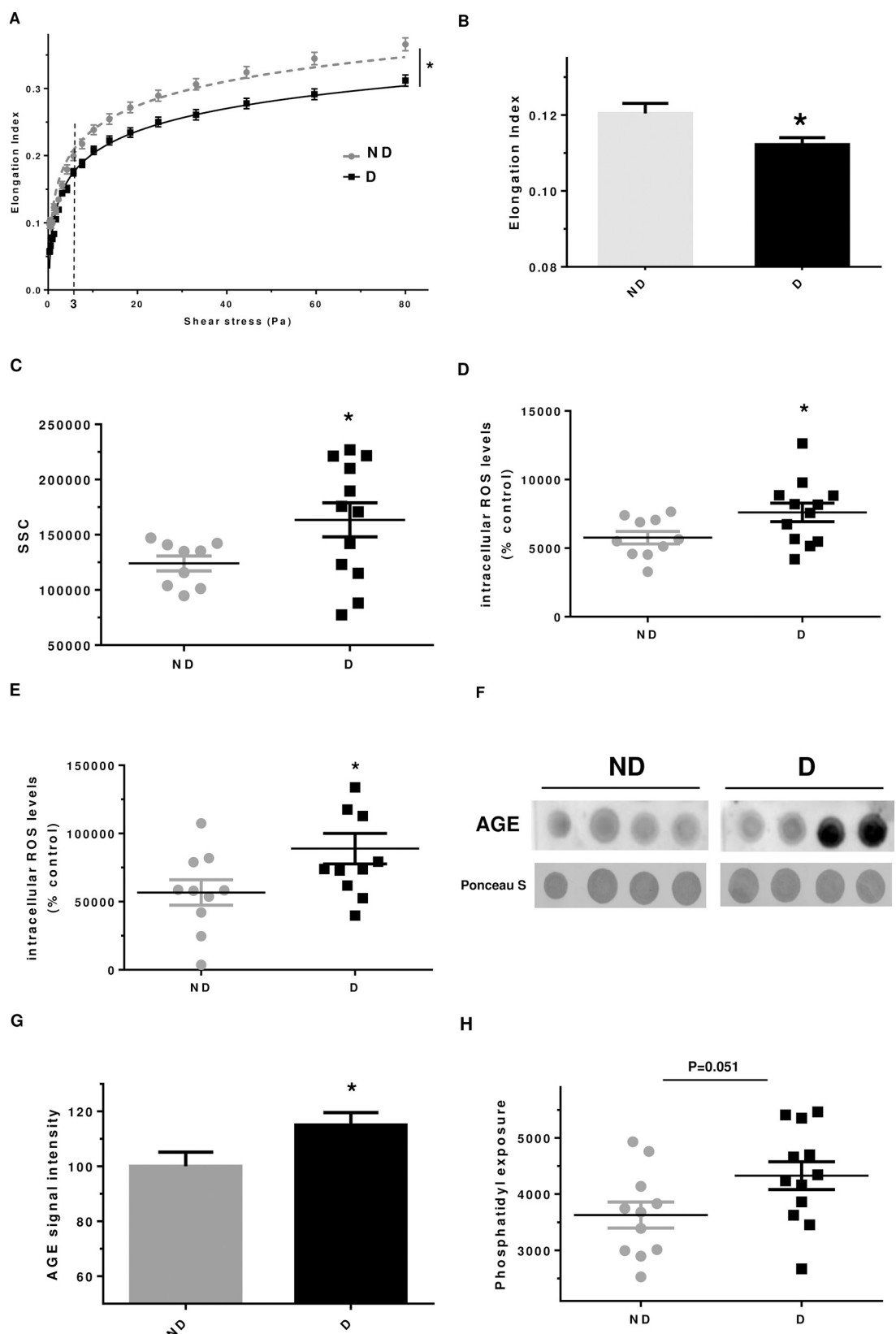

**Fig 5. Erythrocytes from diabetics exhibit altered morphology and enhanced oxidative stress.** (A) Curves correspond to the elongation index of erythrocytes determined by LORRCA measurement as a function of the shear stress intensity (in Pascal unit). (B) Delta elongation index, results are mean ± SEM (n = 9 ND and 12 D), *p<0.05 (Student's t test). (C) Geo mean Side Scatter (SSC) value of erythrocytes analysed by flow cytometry, *p<0.05 (Student's t test). (D) Intracellular ROS formation level in erythrocyte evaluated using DHE probe by cytometry, *p<0.05 (Student's t test). (E) Intracellular ROS formation in erythrocyte preparation was evaluated using DCFDA probe by cytometry. Results are expressed as mean ± SEM, *p<0.05 (Student's t test). (FꞒ) Phosphatidylserine exposure in erythrocytes preparations was quantified by cytometry as described in method section. (G) Representative AGE dot blot performed on lysate preparations of erythrocytes isolated from diabetic and non-diabetic individuals. (HꞒ) AGE quantification by dot blot, n = 9 ND and 12 D, *p<0.05 (Student's t test).

renders erythrocytes less deformable and leads to the alteration of their structure. We demonstrated that glycated erythrocytes produce more intracellular ROS and exhibit an altered redox balance. Furthermore, we highlight a potentially higher phosphatidylserine exposure of erythrocytes when glycated. All these characteristics observed in *in vitro* glycated erythrocytes were confirmed to occur *in vivo* when analysing erythrocytes isolated from diabetic patients.

To obtain an *in vitro* model of glycation, erythrocytes were incubated during 5 days in the absence or presence of increasing glucose concentrations up. HbA1c levels were first evaluated in our different fractions and pathological percentages were obtained for G137 erythrocytes (7%). Hence, our *in vitro* conditions represent an suitable glycation model inducing HbA1c percentages similar to those that can be measured in diabetic patients. Indeed, a HbA1c content of 6.5% corresponds to the threshold that is currently used to diagnose person with diabetes [27]. A more detailed characterization of our preparations at a molecular level, using mass spectrometry, revealed that erythrocytes incubated with 25 or 137 mM of glucose are significantly more glycated in both hemoglobin subunits (α and β) than erythrocytes incubated in the absence of glucose.

Analysis of erythrocyte morphology revealed a higher fragility (facilitated hemolysis), reduced size, and impaired deformability when glycated at 137mM of glucose. Glycated erythrocyte lysis, may constitute a source of oxidative stress through the high iron content in hemoglobin that could be released in the arterial wall vicinity [13]. Our cytometry results show altered erythrocyte size following *in vitro* glycation which is identical to observations made in aged and senescent erythrocytes. Erythrocyte size is a common biomarker used in clinical analysis and is reported as red blood cell width (RDW) [14]. Very recently, a positive association was found between RDW and the severity of coronary artery disease [14]. Ektacytometry is an adequate methodology to assess erythrocyte deformability [26]. Our data indicate that glycation phenomena induced by the hyperglycemic incubations render erythrocytes significantly less deformable and therefore more rigid than erythrocytes incubated under low glycemic conditions. Whilst ektacytometry is a well defined technique for the diagnosis of specific pathology, its use in research is less developed [26]. However, a recent research article published by the group of Pretorius showed a close link between erythrocyte deformability, hemorheology and cardiovascular dysfunction parameters [31]. The impaired capacity of

**Table 4. Intracellular enzymatic activities of erythrocytes isolated from non-diabetic and diabetic.** Catalase, superoxide dismutase, peroxidase and proteasome enzymatic activities were determined as described in the section of material and methods. Results are expressed as mean ± SD (n = 9 ND and 12 D), *P <0.05 *vs*. ND (Student's t-test).

| | ND | | | D | | |
|---|---|---|---|---|---|---|
| **SOD** AU/mg prot | 100 | ± | 25 | 84 | ± | 31 |
| **Catalase** AU/mg prot | 100 | ± | 28 | 100 | ± | 33 |
| **Peroxidase** AU/μg prot | 100 | ± | 14 | 90 | ± | 11* |
| **LLVY** AU/μg prot | 100 | ± | 72 | 161 | ± | 36* |

erythrocytes to deform when glycated may have significant implications in the progression of vascular complications in diabetes. Indeed, *in vivo* glycated erythrocytes may exhibit an altered capacity to pass through tiny vessels like those present in intraplaque neovascularization, contributing to plaque progression and instability [15].

Oxidative stress and damage caused by ROS are implicated in the development of pathologies and in diabetes complications [4,33]. In this study, enhanced ROS formation was observed in glycated erythrocytes, associated with a reduced peroxidase activity. Free radicals and oxidants such as $O_2$˚-, HO˚ and $H_2O_2$, may arise from the high oxygen pressure and from the iron present in hemoglobin [13]. In addition, the generation of reactive oxygen species observed in our glycated erythrocyte model could also result from glucose auto-oxidation [28]. Under our experimental conditions, decreased peroxidase activity lead to reduced $H_2O_2$ "detoxification" by catalysis to $H_2O$. Hydrogen peroxide can generate the highly reactive hydroxyl radical HO˚ through Fenton reaction involving iron [7]. Similar results in term of oxidative stress has been observed in erythrocytes when isolated from insulin resistant obese children [34].

Oxidative stress can lead to the formation of oxidised compounds that may affect protein structure and function [35]. A significant accumulation of the oxidative biomarker 4-HNE was detected in erythrocytes glycated with 25 and 137 mM glucose (Fig 4B and 4C) and associated with a significant reduction in the chymotrypsin-like activity of the proteasome (Table 3). If proteasome activity may be activated under moderate oxidative stress, in higher oxidant conditions a decrease in proteolytic activity may occur [32]. Friguet et al., identified the 4-hydroxynonenal (4-HNE) as a specific oxidant that can inhibit proteasome activity though its binding to the enzymatic protein complex [32]. Under our experimental conditions, the observed significant impairment in proteasome activity in glycated erythrocytes might result from the enhanced ROS formation and 4-HNE content present in our glycated erythrocyte preparations. This reduced proteasome activity may contribute to the altered redox status in G137 erythrocytes leading to increased oxidised protein accumulation. In addition, oxidised proteins that are not degraded by the impaired proteasome system may also contribute to the enhanced ROS generation in G137 erythrocytes. Enhanced 4-HNE accumulation in glycated erythrocytes may induce adduct formation in proteasome subunit leading to proteolytic activity inhibition [36]. If the proteasome plays an important role in controlling redox homeostasis and in degradation oxidised proteins [32], its activities in erythrocytes remain poorly studied. It is worth noting that in a recent study, using a proteomic analysis performed on blood, the 20 S proteasome was identified as a target for glycation in erythrocytes isolated from diabetic patients [37].

Phagocytosis experiments revealed significantly increased phagocytosis of *in vitro* glycated erythrocytes by endothelial cells. This phagocytosis was associated with a higher phosphatidylserine exposure at the surface of glycated erythrocytes attesting their enhanced eryptosis. Abnormal adherence and phagocytosis of erythrocytes by endothelial cells has been described in vascular complications such as atherosclerosis and abdominal aortic aneurysm [13,16].

Most of the types of erythrocyte damage observed with *in vitro* glycated erythrocytes were also observed in erythrocytes isolated from diabetic patients. Indeed, an altered structure associated with enhanced ROS production and modified redox balance was observed in erythrocytes isolated from diabetic patients in comparison to those isolated from non-diabetics. Interestingly, an enhanced carbonylation of erythrocyte membranes were observed in cell isolated from diabetic patients and correlated with the clinical severity of the pathology [38]. Finally, the triggered eryptosis phenomenon observed in erythrocytes isolated from diabetics could lead to increased phagocytosis by endothelial cells *in vivo*.

Whilst more studies are needed to decipher the role of glycation on erythrocyte capacity in vascular dysfunctions linked to diabetes, the study presented here reveals several novel insights with respect to the impact of glycation on erythrocyte structure, morphology, and capacity to be phagocytosed by endothelial with a possible relevance to diabetes.

## Supporting information

**S1 File.**
(DOCX)

**S1 Fig. Characterisation of glycation percentage in the different erythrocyte preparations by mass spectrometry.** Representative figures of the mass spectra obtained in three independent experiments for each incubation condition: 0 (G0), 5 (G5), 25 (G25) and 137 (G137) mmol/l glucose. On each spectrum, four main peaks were obtained corresponding to α-hemoglobin (αHb; 15130 Da), glycated α-hemoglobin (gαHb; 15330 Da), β-hemoglobin (βHb; 15890 Da), glycated β-hemoglobin (gβHb; 16100 Da).
(DOCX)

**S2 Fig. Diabetes alters erythrocyte morphology.** Typical Forward Scatter (FSC) and Side Scatter (SSC) characteristics represented in dot-blot graph obtained by cytometry erythrocytes from non diabetic (left) and diabetic persons (right).
(DOCX)

## Acknowledgments

A special thank you goes to Dr David Wilkinson for helpful discussions and editing this manuscript. Kind helps from Dr. Catherine CETRE-SOSSAH was greatly appreciated by the Authors.

## Author Contributions

**Conceptualization:** Chloé Turpin, Philippe Rondeau, Emmanuel Bourdon.

**Data curation:** Chloé Turpin, Aurélie Catan, Alexis Guerin-Dubourg, Susana B. Bravo, Ezequiel Álvarez, Jean Van Den Elsen, Philippe Rondeau.

**Formal analysis:** Chloé Turpin, Alexis Guerin-Dubourg, Ezequiel Álvarez.

**Investigation:** Chloé Turpin, Aurélie Catan, Alexis Guerin-Dubourg, Xavier Debussche, Ezequiel Álvarez, Philippe Rondeau.

**Methodology:** Chloé Turpin, Aurélie Catan, Alexis Guerin-Dubourg, Xavier Debussche, Susana B. Bravo, Ezequiel Álvarez, Jean Van Den Elsen, Philippe Rondeau, Emmanuel Bourdon.

**Project administration:** Emmanuel Bourdon.

**Resources:** Jean Van Den Elsen.

**Supervision:** Olivier Meilhac, Philippe Rondeau, Emmanuel Bourdon.

**Validation:** Ezequiel Álvarez, Olivier Meilhac, Philippe Rondeau.

**Writing – original draft:** Chloé Turpin, Emmanuel Bourdon.

**Writing – review & editing:** Chloé Turpin, Aurélie Catan, Alexis Guerin-Dubourg, Xavier Debussche, Susana B. Bravo, Ezequiel Álvarez, Jean Van Den Elsen, Olivier Meilhac, Philippe Rondeau, Emmanuel Bourdon.

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
