## [Decision Letter · Decision Letter 0]

22 Jan 2020

PONE-D-19-34544

Enhanced oxidative stress and damage in glycated erythrocytes

PLOS ONE

Dear Dr. Bourdon,

Thank you for submitting your manuscript to PLOS ONE. After careful consideration, we feel that it has merit but does not fully meet PLOS ONE’s publication criteria as it currently stands. Therefore, we invite you to submit a revised version of the manuscript that addresses the points raised by the reviewer. These are major revisions; therefore, we anticipate addressing al the points raised during review process.

We would appreciate receiving your revised manuscript within the next 3 to 6 months. To enhance the reproducibility of your results, we recommend that if applicable you deposit your laboratory protocols in protocols.io, where a protocol can be assigned its own identifier (DOI) such that it can be cited independently in the future. For instructions see: http://journals.plos.org/plosone/s/submission-guidelines#loc-laboratory-protocols

We look forward to receiving your revised manuscript.

Kind regards,

Dinender K Singla

Academic Editor

PLOS ONE

Journal Requirements:

2. Thank you for your ethics statement:

- Centre Hospitalier Universitaire de la Réunion

- ERMIES (NCT01425866)

- data were analyzed anonymously'

Please amend your current ethics statement to confirm that your named institutional review board or ethics committee specifically approved this study.

"This work was supported by the Ministère de l'Enseignement Supérieur et de la Recherche, the

Université de La Réunion and by the European Regional Development Funds RE0001897 (EURégion

Réunion -French State national counterpart).

CT is a recipient of a fellowship grant from the Ministère de l'Education Nationale, de

l'Enseignement Supérieur et de la Recherche, La Réunion University (Contrat Doctoral)."

Reviewers' comments:

Reviewer's Responses to Questions

**Comments to the Author**

1. Is the manuscript technically sound, and do the data support the conclusions?

Reviewer #1: Yes

2. Has the statistical analysis been performed appropriately and rigorously? 

Reviewer #1: Yes

3. Have the authors made all data underlying the findings in their manuscript fully available?

Reviewer #1: Yes

4. Is the manuscript presented in an intelligible fashion and written in standard English?

Reviewer #1: Yes

5. Review Comments to the Author

Reviewer #1: Manuscript ID PONE-19-34544 presents interesting data on the effects of glycation on measures of erythrocyte structure, oxidative state and uptake/breakdown by endothelial cells. In general the manuscript is well-written.

Major Comments:

-Have the antibodies been validated for the dot blots and western blots? Were positive or negative controls performed?

-It is interesting that the western blots showed an initial increase in AGEs following incubation with G5 and G25, but not with G137 whereas the dot blots showed significantly higher AGEs at G25 and G137.

-Line 247: These HT50 data are shown in figure 1C, not figure 3. It is interesting that the HT50 trend is no longer seen with the G137 dose. Might that be related to the loss of AGE increase at the highest dose as well?

-Discussion material (references) should not appear in the results section of a manuscript. As currently written, the results section reads more like a combined results and discussion section. Justification for the various assays could also be moved to the methods describing them.

-Line 277 mentions a progressive increase in DCF fluorescence. However, only the highest dose of glucose produced significantly greater fluorescence than G0. Therefore, I do not think one can conclude that the fluorescence was progressively increased.

-Line 288: mentions that peroxidase activity was significantly reduced in G137 cells compared to G25. However, the table indicates a significant difference between G137 and G0 cells.

-Figure S6 seems to show variability in the amount of glycation within the diabetic samples with only 2 showing increased glycation. Were these samples from 4 different subjects?

-Line 330 mentions a significant reduction in SOD in the diabetic samples. However, this significance is not shown in table 4.

-Lines 336-337: mentions that the phosphatidylserine exposure was significantly higher in diabetic samples. This conclusion cannot be made as the p-value was not significant (0.051). Moreover, the conclusion regarding this variable in the discussion is likewise an overstatement (lines 349-351).

-It is interesting that the in vitro exposed cells showed reduced LLVY with glycation whereas this variable was increased in the diabetic samples. What could explain this difference? It appears the in vitro experiments do not accurately model this variable.

Minor Comments:

-Lines 100, 224: Did you mean “fructosamine” as opposed to “frutosamine”?

-Inter- and intra-assay coefficients were provided for the free radical-induced hemolysis test, but not for other assays.

-Have the SOD and catalase activity assays been previously validated? (ie. Provide citations)

-What is meant by “fixed erythrocytes”, line 168? Do you mean affixed to the plate?

-Table 1 and figures should be able to stand alone from a paper. Please define G0, G5, G25, and G137 in the table legend.

-Figure 2: The labelling is confusing as the group of 4 figures do not need separate A-D labels. Consider labeling A-D, simply “A”.

-Line 281-282: It is mentioned that 4-HNE was higher in G25 and G137 erythrocytes compared to G5. Were levels not also higher than G0?

-lines 362-363: “glycated erythrocyte lysis when glycated”??

-Line 373: delete “Pr”

6. PLOS authors have the option to publish the peer review history of their article (what does this mean?). If published, this will include your full peer review and any attached files.

Reviewer #1: No

---

## [Author Response · Author response to Decision Letter 0]

6 Mar 2020

Academic Editor

1. Please ensure that your manuscript meets PLOS ONE's style requirements.

We did ensure our manuscript meets PLOS ONE's style requirements including those for file naming.

2. Ethics statement.

We did amend our current ethics statement to confirm that our study was specifically approved by the named institutional review board or ethics committee in the Methods section of our manuscript. Also, we added the same text to the “Ethics Statement” field of the submission form (via “Edit Submission”).

3. Acknowledgments Section.

Any funding-related text from the manuscript was removed.

Our funding statement was updated:

"This work was supported by the Ministère de l'Enseignement Supérieur et de la Recherche, the Université de La Réunion, the "Structure fédérative de recherche biosécurité en milieu tropical (BIOST) and by the European Regional Development Funds RE0001897 (EU- Région Réunion -French State national counterpart)."

4 PLOS ONE Policy on Figure Preparation

All six images displayed in our MS (figures 1A, 3B, S3A and S6) were prepared in accordance with PLOS ONE Policy on Figure Preparation:

They were not be adjusted in any way that could affect the scientific information displayed.

Images in figures are not overcropped around the bands of interest.

All relevant samples for comparative analysis were run on the same gel/blot.

In the figures of our MS there are no composite images of bands originating from different blots.

5. The phrase “data not shown” 

The phrase “data not shown” has been removed in the revised version of our manuscript.

6. Supporting Information

Captions for our Supporting Information files were included at the end of your manuscript, and any in-text citations were updated to match accordingly.

 Reviewer #1: 

Major Comments:

-Have the antibodies been validated for the dot blots and western blots? Were positive or negative controls performed?

Antibodies used in this study, anti-AGE (ab23722) and anti-4-HNE (ab46545) have been validated for western blots and also for ELISA and immunohistochemistry. Positive controls were performed. For 4-HNE and AGE dot blots, oxidized low density lipoproteins (ox LDL) and glycated albumin (GA) were used as positive controls, respectively.

Here are examples of AGE and 4-HNE dot blots including positive controls:(photos are included in the cover letter)

-It is interesting that the western blots showed an initial increase in AGEs following incubation with G5 and G25, but not with G137 whereas the dot blots showed significantly higher AGEs at G25 and G137.

Yes we agree, AGE evaluation by using western blot and dot blot techniques, evidenced increase AGE accumulation in glycated erythrocytes compared to control (G0 erythrocytes).

If significant increase AGE accumulation was measured by using dot blot for G25 and G137 compared with the control (Fig 1), the increase was significant when using western blot with G5 and G25, but not with G137 compared to the control (Fig S3). This slight difference in terms of statistical difference may arise from a more accurate quantification of dot blot signal than the signal quantification on western blot. Results in term of significant increase in AGE accumulation in our glycated erythrocyte preparations were comforted by different techniques, dot blot, western blot, mass spectrometry and also by the determination of glycated hemoglobin (% HbA1c), the 5-hydroxymethylfurfural and the early glycation product (EGP) levels. 

-Line 247: These HT50 data are shown in figure 1C, not figure 3. It is interesting that the HT50 trend is no longer seen with the G137 dose. Might that be related to the loss of AGE increase at the highest dose as well?

Thank you for pointing this out to us. The figure number has been corrected.

We agree, HT50 trend is no longer seen with the G137 dose. As previously written, significant increases in AGE accumulation in our glycated erythrocyte preparation were comforted by different techniques, dot blot, western blot, mass spectrometry and also by the determination of glycated hemoglobin (% HbA1c), the 5-hydroxymethylfurfural and the early glycation product (EGP) levels. All data obtained, but western blot quantification, evidenced significant AGE accumulation increase for G137 samples compared to the control.

Hence, we would rather think that this lower capacity of G137 sample to resist to a free radical-induced hemolysis may come from AGE accumulation. 

-Discussion material (references) should not appear in the results section of a manuscript. As currently written, the results section reads more like a combined results and discussion section. Justification for the various assays could also be moved to the methods describing them.

As suggested discussion material including references were moved to the method section.

-Line 277 mentions a progressive increase in DCF fluorescence. However, only the highest dose of glucose produced significantly greater fluorescence than G0. Therefore, I do not think one can conclude that the fluorescence was progressively increased.

We agree, we modified our text and removed the term "progressive increase".

-Line 288: mentions that peroxidase activity was significantly reduced in G137 cells compared to G25. However, the table indicates a significant difference between G137 and G0 cells.

Thank you for pointing this out to us, text was corrected.

-Figure S6 seems to show variability in the amount of glycation within the diabetic samples with only 2 showing increased glycation. Were these samples from 4 different subjects?

Yes, these samples were from 4 different subjects, 4 diabetic and 4 non-diabetic individuals.

If variability was observed in the amount of glycation within the diabetic samples (Fig S6), AGE quantification revealed a significantly higher accumulation in erythrocytes from diabetics compared to erythrocytes from non-diabetic individuals (Fig 4E). 

-Line 330 mentions a significant reduction in SOD in the diabetic samples. However, this significance is not shown in table 4.

Thank you, our text was modified:

"Although catalase and SOD activities appeared not to be significantly different between both groups (ND vs. D), a significant reduced peroxidase activity was measured in erythrocytes isolated from diabetic persons (table 4). This reduction in peroxidase activity, associated with high glucose concentration, could explain the enhanced ROS formation in erythrocytes isolated from diabetic persons."

-Lines 336-337: mentions that the phosphatidylserine exposure was significantly higher in diabetic samples. This conclusion cannot be made as the p-value was not significant (0.051). Moreover, the conclusion regarding this variable in the discussion is likewise an overstatement (lines 349-351).

We agree and therefore changed our text. Line 336 337 we wrote "Erythrocytes from diabetic persons exhibit a tendency for a higher phosphatidylserine exposure than erythrocytes isolated from non-diabetic persons."

Lane 351 we wrote " Furthermore, we highlight a potentially higher phosphatidylserine exposure of erythrocytes when glycated."

-It is interesting that the in vitro exposed cells showed reduced LLVY with glycation whereas this variable was increased in the diabetic samples. What could explain this difference? It appears the in vitro experiments do not accurately model this variable.

Yes, this is indeed very interesting!

We developed on this in the revised version of our MS. Briefly, if proteasome activity may be activated under moderate oxidative stress, in higher oxidant conditions a decrease in proteolytic activity may occur [1]. Friguet et al., identified the 4-hydroxynonenal (4-HNE) as a specific oxidant that can inhibit proteasome activity though its binding to the enzymatic protein complex [1].

In vitro, significant reduction in the chymotrypsin-like activity of the proteasome (LLVY) was associated with a significant accumulation of the oxidative biomarker 4-HNE in erythrocytes glycated with 25 and 137 mM glucose. In vivo, if oxidative stress was higher in erythrocytes from diabetics than from non-diabetics, no variation was measured in term of 4-HNE accumulation. We think that enhanced oxidative stress in in vivo glycated erythrocytes if significant may be moderate enough to trigger proteasome LLVY activation.

Minor Comments:

-Lines 100, 224: Did you mean “fructosamine” as opposed to “frutosamine”?

The text was corrected. Fructosamine correct term was used.

-Inter- and intra-assay coefficients were provided for the free radical-induced hemolysis test, but not for other assays.

Yes, all assays used in our study were validated with citations provided and we keep at heart to give maximum information about our techniques such as inter and intra coefficient assay, which we only have for the free radical-induced hemolysis test as it was developed and patented in my former research group [2]. This later citation is provided in the revised version of our MS.

-Have the SOD and catalase activity assays been previously validated? (ie. Provide citations)

Yes, SOD and catalase activity assays were previously validated. A citation of a very recent work published by our group was included in the new revised version of our MS: [3]

-What is meant by “fixed erythrocytes”, line 168? Do you mean affixed to the plate?

After 24-hours of incubation with erythrocytes, endothelial cells were washed 3 times with PBS and treated with water for 3 minutes to induce lysis (by hypotonic shock) of any fixed erythrocytes at the cellular membrane surface and supernatant was discarded.

This text was included in the new version of our revised MS.

-Table 1 and figures should be able to stand alone from a paper. Please define G0, G5, G25, and G137 in the table legend.

This is an excellent suggestion. Our different erythrocyte preparations G0, G5, G25, and G137 are now defined in the table and figure legend in the revised version of our MS.

-Figure 2: The labelling is confusing as the group of 4 figures do not need separate A-D labels. Consider labeling A-D, simply “A”.

We agree. Figure labelling was changed. Previous figure 2AD is now figure 2A and previous figure 2E is now figure 2B.

-Line 281-282: It is mentioned that 4-HNE was higher in G25 and G137 erythrocytes compared to G5. Were levels not also higher than G0?

Actually 4-HNE levels were significantly higher in G25 and G137 erythrocytes compared to G5.

-lines 362-363: “glycated erythrocyte lysis when glycated”??

Thank you, our text was corrected.

-Line 373: delete “Pr”

Text was corrected.

References:

1. Friguet B (2006) Oxidized protein degradation and repair in ageing and oxidative stress. FEBS Lett 580: 2910-2916.

2. Prost M (1992) Process for the determination by means of free radicals of the antioxidant properties of a living organism or a potentially aggressive age. United States.Patent 5.135.850. Aug 4

3. Dobi A, Bravo SB, Veeren B, Paradela-Dobarro B, Alvarez E, et al. (2019) Advanced glycation end-products disrupt human endothelial cells redox homeostasis: new insights into reactive oxygen species production. Free Radic Res 53: 150-169.

---

## [Decision Letter · Decision Letter 1]

5 May 2020

PONE-D-19-34544R1

Enhanced oxidative stress and damage in glycated erythrocytes

PLOS ONE

Dear Dr. Bourdon,

Thank you for submitting your revised manuscript to PLOS ONE. After careful consideration, we feel that it has merit but does not fully meet PLOS ONE’s publication criteria as it currently stands. These are minor revisions. Therefore, we invite you to submit a revised version of the manuscript that addresses the points raised during the review process.

We would appreciate receiving your revised manuscript by Jun 19 2020 11:59PM. To enhance the reproducibility of your results, we recommend that if applicable you deposit your laboratory protocols in protocols.io, where a protocol can be assigned its own identifier (DOI) such that it can be cited independently in the future. For instructions see: http://journals.plos.org/plosone/s/submission-guidelines#loc-laboratory-protocols

We look forward to receiving your revised manuscript.

Kind regards,

Ping Song, Ph.D

Academic Editor

PLOS ONE

Reviewers' comments:

Reviewer's Responses to Questions

**Comments to the Author**

1. If the authors have adequately addressed your comments raised in a previous round of review and you feel that this manuscript is now acceptable for publication, you may indicate that here to bypass the “Comments to the Author” section, enter your conflict of interest statement in the “Confidential to Editor” section, and submit your "Accept" recommendation.

Reviewer #1: All comments have been addressed

Reviewer #2: (No Response)

2. Is the manuscript technically sound, and do the data support the conclusions?

Reviewer #1: Yes

Reviewer #2: Yes

3. Has the statistical analysis been performed appropriately and rigorously? 

Reviewer #1: Yes

Reviewer #2: No

4. Have the authors made all data underlying the findings in their manuscript fully available?

Reviewer #1: Yes

Reviewer #2: No

5. Is the manuscript presented in an intelligible fashion and written in standard English?

Reviewer #1: Yes

Reviewer #2: Yes

6. Review Comments to the Author

Reviewer #1: (No Response)

Reviewer #2: The highlight of this manuscript is relatively comprehensive comparison on the structure change, redox status and the EC phagocytosis of erythrocytes under the condition of glycation between in vivo and invitro. But there are still some parts needed to be improved and further explained. Some of the results showed opposite phonotype in in vitro and in vivo, which should be well discussed.

Majors:

In the hemolysis study, the results indicate that low concentrated glucose incubation had a protective membrane-stabilizing effect, which means an increased “flexibility” of the cell membrane or less fragile membrane. However, in the test of deformability, elongation index was reduced after glucose incubation, which suggests a reduced “flexibility” of the cell membrane and/or cytoskeleton. How does the authors explain these possible “conflict” phonotypes?

In some papers (for example PMID: 31704555), erythrocytes from diabetes showed impaired hemolysis resistance, which is opposite the phenomenon observed in this manuscript. Do the in vitro and in vivo glycations lead to different hemolysis results?

The high glucose concentration may change the osmolarity of the solution and making the solution hypertonic, therefore the cells may already in a shrank state in the initial stage, which may cause influence on the results. Did the authors detect the cell morphology with microscopy?

The proteasome activity detected in in vitro and in vivo showed opposite results in the manuscript. When treated with high glucose level, impaired proteasome activity was detected. But the proteasome activity from the diabetic erythrocytes showed an increased activity. How does the author explain these results?

Minors:

The data analysis used in Table2 & 3 should be One-way ANOVA.

If possible, can the author add the AOPP data back?

In the method section, water was used to induce lysis of the erythrocytes sticking on the surface of ECs. EC adhesion is not very strong. Does this method also wash away some EC and affect the quantification of internalized erythrocytes?

The figures need to be reorganized. The supplementary figures can be integrated in regular figure as well.

7. PLOS authors have the option to publish the peer review history of their article (what does this mean?). If published, this will include your full peer review and any attached files.

Reviewer #1: No

Reviewer #2: No

---

## [Author Response · Author response to Decision Letter 1]

5 Jun 2020

Majors:

In the hemolysis study, the results indicate that low concentrated glucose incubation had a protective membrane-stabilizing effect, which means an increased “flexibility” of the cell membrane or less fragile membrane. However, in the test of deformability, elongation index was reduced after glucose incubation, which suggests a reduced “flexibility” of the cell membrane and/or cytoskeleton. How does the authors explain these possible “conflict” phonotypes?

It is true, erythrocytes incubated with low glucose concentration exhibit an enhanced resistance to free radical-induced hemolysis concomitantly with an impaired flexibility of their membrane. Then, results obtained in the hemolysis and deformability tests may appear conflictual. Actually, these two tests did not assess the same biological phenomenon. Our in vitro free radical-induced blood hemolysis test allows determination of erythrocyte capacity to resist to an oxidative stress mediated by a free radical generator (AAPH), a chemical agent. Results were expressed as the half time of hemolysis (HT50) and reported on figure 2E.

The deformability of the different erythrocyte preparations was analysed by ektacytometry. This methodology allows elongation index determination, a biophysical index which expresses erythrocyte deformability capacity in response to increasing shear stress (Fig 2F and 2G).

Hence, if our “hemolysis test” measures erythrocyte capacity to resist to hemolysis when submitted to a free radical generator, elongation index specifically expresses membrane flexibility/rigidity. In other words, these two different methods give two types of results.

Concerning results, erythrocytes that have been incubated with high glucose concentrations exhibit both impaired hemolysis resistance capacity (2E) and reduced membrane flexibility (Fig 2F and 2G).

In some papers (for example PMID: 31704555), erythrocytes from diabetes showed impaired hemolysis resistance, which is opposite the phenomenon observed in this manuscript. Do the in vitro and in vivo glycations lead to different hemolysis results?

Thank you for mentioning this.

We agree, in some papers, erythrocytes from diabetes showed impaired hemolysis resistance. Please note, the mentioned reference (PMID: 31704555) is a recent article from our group. In this article, erythrocytes from diabetic patients exhibited a slight non-significant decrease in free radical-induced hemolysis half time in comparison with erythrocytes from non-diabetic (Cf fig A).

Figure 1 is issued from Catan et al (2019) Atherosclerosis.

Considering phenomenon observed in our present manuscript, if erythrocytes in vitro incubated with 5 or 25 mM of glucose (G5, G25) exhibited a higher HT50 compared to G0 (Figure 2E), those incubated with 137 mM of glucose (G137) do not exhibit such hemolysis resistance any more. In addition, HbA1c measurement in our in vitro glycated preparations revealed that values obtained for G137 erythrocytes (7%) were highly similar to HbA1c values observed in diabetic patients (table 1). 

Hence, in our experimental conditions, in vitro and in vivo glycation lead to similar hemolysis results.

Finally, please note in our present study, a slight non-significant impairment in HT50 for erythrocytes was also observed when isolated from diabetics when compared to non-diabetic (Cf figure 2). 

Figure 2. This figure was established with data from the present study.

The high glucose concentration may change the osmolarity of the solution and making the solution hypertonic, therefore the cells may already in a shrank state in the initial stage, which may cause influence on the results. Did the authors detect the cell morphology with microscopy?

It is true, high glucose concentration may change the osmolarity of the solution making cell potentially in a shrank state in the initial stage.

In our experimental conditions, we did not observe strong evidence of such phenomenon by microscopy (fig 3). If an increased number erythrocyte ghosts is observed when incubated with increasing glucose concentrations (see arrows), most erythrocytes does not exhibit a significant shrank state in relation to glucose concentration.

In addition, if such phenomenon would occur, please note our erythrocyte preparations used in the present MS, were washed in an isotonic solution prior analysis. 

Figure 3. This figure corresponds to erythrocyte preparations that were incubated with 5 (A), 50 (B) or 100 (C) mM glucose for five days. Please note these preparations were not washed in an isotonic solution prior microscopic analysis. Bar corresponds to 10 µm.

The proteasome activity detected in in vitro and in vivo showed opposite results in the manuscript. When treated with high glucose level, impaired proteasome activity was detected. But the proteasome activity from the diabetic erythrocytes showed an increased activity. How does the author explain these results?

We agree with this and explanations are provided in our revised version of our MS.

Page 14 we wrote:

“Conversely, the chymotrypsin-like activity of the proteasome appeared to be significantly enhanced in erythrocytes purified from diabetics compared to those from non-diabetics (table 4). Proteasome activity may be activated under moderate oxidative stress [1]. In vivo, if oxidative stress is higher in erythrocytes from diabetics than from non-diabetics, no variation was measured in term of 4-HNE accumulation. Enhanced oxidative stress in in vivo glycated erythrocytes may be significant and moderate enough to trigger proteasome LLVY activation.” 

Minors:

The data analysis used in Table2 & 3 should be One-way ANOVA.

We followed Reviewer’s advice and used One-way ANOVA for statistical analysis in table 2 and 3.

Modifications in the revised version of our MS were reported lines 236, 287 and 289.

If possible, can the author add the AOPP data back?

We followed Reviewer’s advice, page 14 we wrote “In addition, accumulation of advanced oxidation product (AOPP), was found to be higher (+18%) in erythrocytes from diabetics but this increase did not reach significance.”

In the method section, water was used to induce lysis of the erythrocytes sticking on the surface of ECs. EC adhesion is not very strong. Does this method also wash away some EC and affect the quantification of internalized erythrocytes?

Yes, we did use water to induce lysis of erythrocytes sticking on the surface of cultured endothelial cells. This method may indeed wash away some endothelial cells and such phenomenon was noticed when cells were treated with highly glycated erythrocytes. Still, this phenomenon could not affect our results since enhanced internalization of erythrocytes was significant and observed in endothelial cells that have been incubated with glycated erythrocytes. In addition, please note erythrocyte phagocytosis by endothelial cells was confirmed by recent unpublished data using flow cytometry and confocal microscopy. These results would be part of a future MS.

The figures need to be reorganized. The supplementary figures can be integrated in regular figure as well.

 According to Reviewer’s advice, our figures were reorganized in the new version of our MS.

Four previous supplementary figures are now integrated in regular figures in the present version of our MS.

It concerns:

- Figure #1 (previous figure S1): Early glycation product detection by using fluorescent boronic acids.

- Figure 2C and 2D (previous figure S3): Enhanced AGE formation in glycated erythrocytes.

- Figure 5E (previous figure S5): Enhanced oxidative stress in erythrocytes from diabetics.

- Figure 5G (previous figure S6): Enhanced AGE content in erythrocytes from diabetics.

[1] Friguet, B. Oxidized protein degradation and repair in ageing and oxidative stress. FEBS Lett 580:2910-2916; 2006.

---

## [Decision Letter · Decision Letter 2]

15 Jun 2020

Enhanced oxidative stress and damage in glycated erythrocytes

PONE-D-19-34544R2

Dear Dr. Bourdon,

We’re pleased to inform you that your manuscript has been judged scientifically suitable for publication and will be formally accepted for publication once it meets all outstanding technical requirements.

Kind regards,

Ping Song, Ph.D

Academic Editor

PLOS ONE

Additional Editor Comments (optional):

The authors have addressed the concerns raised by two reviewers. The manuscript is acceptable in PLOS ONE.

Reviewers' comments:

Reviewer's Responses to Questions

**Comments to the Author**

1. If the authors have adequately addressed your comments raised in a previous round of review and you feel that this manuscript is now acceptable for publication, you may indicate that here to bypass the “Comments to the Author” section, enter your conflict of interest statement in the “Confidential to Editor” section, and submit your "Accept" recommendation.

Reviewer #2: All comments have been addressed

2. Is the manuscript technically sound, and do the data support the conclusions?

Reviewer #2: Yes

3. Has the statistical analysis been performed appropriately and rigorously? 

Reviewer #2: Yes

4. Have the authors made all data underlying the findings in their manuscript fully available?

Reviewer #2: Yes

5. Is the manuscript presented in an intelligible fashion and written in standard English?

Reviewer #2: Yes

6. Review Comments to the Author

Reviewer #2: They authors have addressed all the comments. But the figures still need reorganization to be appropriated displayed in a printed page. A single figure should is composed of multiple well-arranged graphs. The authors' figures still display as individual graph which does not meet the requirement of publication. Please look at the figure pattern in other published PLOS papers.

7. PLOS authors have the option to publish the peer review history of their article (what does this mean?). If published, this will include your full peer review and any attached files.

Reviewer #2: No

---

## [Editor Report · Acceptance letter]

17 Jun 2020

PONE-D-19-34544R2 

Enhanced oxidative stress and damage in glycated erythrocytes 

Dear Dr. Bourdon:

I'm pleased to inform you that your manuscript has been deemed suitable for publication in PLOS ONE. Congratulations! Your manuscript is now with our production department. 

Kind regards, 

on behalf of

Dr. Ping Song 

Academic Editor

PLOS ONE